# Direct Sarcomere Modulators Are Promising New Treatments for Cardiomyopathies

**DOI:** 10.3390/ijms21010226

**Published:** 2019-12-28

**Authors:** Osamu Tsukamoto

**Affiliations:** 1Department of Medical Biochemistry, Osaka University Graduate School of Medicine, 2-2 Yamadaoka, Suita 565-0871, Japan; tsuka@medbio.med.osaka-u.ac.jp; Tel.: +81-6-6879-3492; 2Department of Medical Biochemistry, Graduate School of Frontier Bioscience, Osaka University, 1-1 Yamadaoka, Suita 565-0871, Japan

**Keywords:** sarcomere, direct sarcomere modulators, hypertrophic cardiomyopathy, dilated cardiomyopathy

## Abstract

Mutations in sarcomere genes can cause both hypertrophic cardiomyopathy (HCM) and dilated cardiomyopathy (DCM). However, the complex genotype-phenotype relationships in pathophysiology of cardiomyopathies by gene or mutation location are not fully understood. In addition, it is still unclear how mutations within same molecule result in different clinical phenotypes such as HCM and DCM. To clarify how the initial functional insult caused by a subtle change in one protein component of the sarcomere with a given mutation is critical for the development of proper effective treatments for cardiomyopathies. Fortunately, recent technological advances and the development of direct sarcomere modulators have provided a more detailed understanding of the molecular mechanisms that govern the effects of specific mutations. The direct inhibition of sarcomere contractility may be able to suppress the development and progression of HCM with hypercontractile mutations and improve clinical parameters in patients with HCM. On the other hand, direct activation of sarcomere contractility appears to exert unexpected beneficial effects such as reverse remodeling and lower heart rate without increasing adverse cardiovascular events in patients with systolic heart failure due to DCM. Direct sarcomere modulators that can positively influence the natural history of cardiomyopathies represent promising treatment options.

## 1. Introduction

Sarcomeres are the fundamental contractile units of cardiomyocytes. Mutations in a number of proteins associated with sarcomeres lead to various forms of cardiomyopathy such as hypertrophic cardiomyopathy (HCM) and dilated cardiomyopathy (DCM). HCM is the most common genetic heart disease and a major cause of cardiac sudden death and heart failure in young people. HCM is characterized by unexplained myocardial hypertrophy accompanied by myofibrillar disarray and diastolic dysfunction. Pathogenic variants of almost all proteins in the sarcomere have been detected in HCM, which is commonly defined as a disease of the sarcomere. However, the exact mechanisms of HCM pathogenesis remain unclear. Thus, there are currently no effective drugs to treat HCM. By contrast, DCM is characterized by the dilation of the left ventricle with impaired systolic function, which is a leading cause of heart failure with reduced ejection fraction (HFrEF), left ventricular assist device implantation, and heart transplantation. Compared with HCM, DCM has much greater locus heterogeneity. More than 40 different genes have been implicated in DCM [1]. These genes encode proteins in the sarcomere and nuclear lamina, desmosome, intermediate filaments, and dystrophin-associated glycoprotein complex. Diminished sarcomere force generation, impaired force transmission, alternations in energy production, defects in calcium handling, and other mechanisms have been proposed as molecular mechanisms of DCM pathogenesis depending on the affected genes. There has been substantial progress in drug therapy for the management of HFrEF, such as β-adrenergic blockers. However, drug therapy is of limited effectiveness for patients with advanced DCM because traditional inotropic agents improve hemodynamic status at the expense of an increased risk of mortality.

Fortunately, recent technological advances have made it possible to identify the initial event caused by pathogenic mutations in cardiomyopathies, which can lead to the development of new effective drugs. In this review are outlined the recent major advances in the understanding of mechanisms underlying cardiomyopathies caused by mutations in sarcomere proteins and new drugs that target sarcomere proteins.

## 2. Ensemble Force and Sarcomere Power Output 

The power output of the heart, defined as the product of the pressure times flow (volume per unit time), is an important hemodynamic parameter reflecting the ability of the heart to efficiently pump blood. Sarcomeres are composed of thick and thin filaments and the fundamental units of contraction in striated muscles (Figure 1a). Each myosin head works as an independent force generator with its own intrinsic force (*f*). Importantly, the actin-activated chemomechanical cycle is divided into two fundamental parts: The weakly bound and strongly bound states of the myosin heads (Figure 1b) [2,3]. The key step in the transition from the weakly bound state to the strongly bound force-generating state is the release of inorganic phosphate from myosin heads [4]. Phosphate release rapidly induces closure of the actin-biding cleft, which is coupled to the large movement of the myosin lever arm (power stroke) [4]. Then, an additional final component of the lever arm swing occurs, which releases MgADP from myosin heads [4]. Release of ADP is rapidly followed by the binding of MgATP, which leads to the transition from the strongly bound state to the weakly bound state by dissociating myosin heads from actin filament [4]. Since only myosin heads in the strongly bound state can produce force, duty ratio is defined as the fraction of myosin heads in sarcomeres that are strongly bound to actin and generating force. Duty ratio is expressed as *t_s_*/*t_c_*, where *t_s_* is the strongly bound state time and *t_c_* is the total cycle time of the actin-activated myosin chemomechanical cycle, expressed as
*t_c_* = 1/*k_cat_*(1)

Accordingly, Spudich [2,3] referred to the total force produced by cardiac muscle as ensemble force (*Fe*), which is expressed as a product of *f* and the number of myosin heads that are bound in a force-producing state,
*Fe* = *f* × (*t_s_*/*t_c_*) × *N_t_*(2)
where *N_t_* is the total number of functionally available myosin heads (Figure 1b). Furthermore, velocity is the distance moved per unit of time in the strongly bound state, expressed as
*v* = *d*/*t_s_*(3)
where *d* is the displacement caused by myosin head stroke amplitude. Thus, at the sarcomere level, sarcomere power (*P*) output is the product of ensemble force generated by myosin heads times velocity of movement along actin filaments, which can be obtained from the force–velocity (F–V) curve of muscle contraction as [2,3]
*P* = *Fe**v*(4)

By examining these individual parameters in each HCM or DCM mutation, we can investigate the pathogenesis of cardiomyopathy at the molecular level. Indeed, several recent studies revealed the molecular mechanism how the cardiomyopathy-causing mutations in sarcomeric proteins affect the force production. R453C mutation in β-cardiac myosin heavy chain (*MYH7*) is a HCM-causing mutation [5]. A dual-beam optical trap, that can quantify the intrinsic force (*f*) generated by single myosin molecule, revealed that R453C mutation increaed the intrinsic force (*f*) [5]. Furthermore, the loaded in vitro motility assay using α-actinin as as a frictional load confirmed that R453C mutation has a higher force-generationg ability than wild type [5]. Thus, the molecular mechanism how R453C mutation causes HCM is considered as a gain of hypercontractile state in the heart muscle by increaed enseble force due to higher intrinsic force (*f*) [5]. D239N mutation in *MYH7* also causes HCM by a gain of hypercontractility due to higher intrinsic force (*f*) [6]. On the other hand, DCM-causing S532P muation in *MYH7* has a signigicanlty depressed maximum velocity and lower ensemble force than wild type [7]. In the case of HCM-causing H251N mutation in *MYH7*, not only intrinsic force (*f*), but also increaesd number of available myosin heads (*N_t_*) appear to contribute to the hypercontractility [6]. The myosin mesa is a putative binding site for the proximal portion of the α-helical coilded-coil tail of myosin S2 domain, which folds back and sequesters myosin heads to limit the number of available myosin heads (*N_t_*). H251N mutaiton is located on the surface of the myosin mesa and disrupts the sequestered state of myosin heads, which leads to increase the number of available myosin heads (*N_t_*) [6]. Recently, the sequestered state of myosin heads have gotten a lot of attention because the relaxation states of myosin dimers in striated muscles have been proved to be important for the development of cardiomyopathy.

## 3. Super-Relaxed State in the Heart

Dynamic and asymmetric interaction of the two heads in a pair of myosin II molecules forms a myosin interacting-heads motif (IHM) structure. During relaxation, there are two relaxation states of myosin dimers in striated muscles, including human cardiac muscle, depending on IHM interaction: The disordered relaxed (DRX) state and the ordered relaxed or super-relaxed (SRX) state [8,9] (Figure 2). In the DRX state, the blocked head (BH) docked onto its S2 and actin-binding interface is positioned on the converter domain of the partner free head (FH). Accordingly, ATPase activity and the binding of BH to actin are inhibited (Figure 2). By contrast, the FH can sway within the inter-filament space toward the actin filament, hydrolyze ATP, and bind to actin, although its binding is restricted by the troponin–tropomyosin complex on the thin filament (Figure 2). In the SRX state, the FH docks onto the BH. Its ATPase activity is highly self-inhibited, resulting in the energy-saving state with minimal ATP consumption. Because the transition from the SRX state to the DRX state increases ATPase activity and subsequently the metabolic rate of skeletal muscle, SRX state destabilization has been considered a potential therapeutic target for obesity [9]. Recently, changes in DRX and SRX states due to mutations in sarcomere proteins in cardiomyopathy have provided new fundamental insights into the molecular pathogenesis of HCM [9,10] and DCM [11]. Alamo et al. characterized the sites of interactions between adjacent myosin heads and associated protein partners in rare variants of cardiomyopathy in 6112 patients with HCM and 13,415 patients with DCM using a human β-cardiac myosin IHM quasi-atomic model [11]. They demonstrated that HCM variants are highly enriched in the converter and mesa domains, both of which participate in IHM interactions, impair IHM formation, destabilize the SRX state, and increase the proportion of myosin in the DRX state [9]. This can account for the characteristics of HCM pathophysiology, such as enhanced contractility, reduced diastolic relaxation, and increased energy consumption. On the other hand, DCM variants were enriched in the nucleotide-binding site, but not in the regions related to IHM interactions [8].

## 4. Hypertrophic Cardiomyopathy (HCM)

HCM-causing mutations have been detected mostly in the genes of sarcomere proteins. Mutations in cardiac myosin binding protein C (*MYBPC3)* and β-cardiac myosin heavy chain (*MYH7*) account for most of cases with identifiable pathogenic variants [12]. Mutations in other sarcomere genes include troponin T (*TNNT2*), troponin I (*TNNI3*), troponin C (*TNNC1*), tropomyosin α-1 chain (*TPM1*), cardiac actin (*ACTC1*), ventricular myosin light chain 2 (*MYL2*), and myosin light chain 3 (*MYL3*). Although many mutations in sarcomere genes have been identified as mutations that cause HCM, it remains unclear how mutations in a single protein lead to an HCM phenotype. There are currently two hypotheses about the molecular mechanisms of HCM pathogenesis.

### 4.1. Hypocontractile Hypothesis

One proposed mechanism of HCM pathogenesis is sarcomere hypocontractility primarily caused by sarcomere gene mutations as the initiating event that subsequently triggers compensatory hypertrophy and remodeling [13]. Witjas-Paalberends et al. demonstrated a consistent significant decrease in maximal force generation capacity in cardiomyocytes and myofibrils from HCM patients with sarcomere mutations in *MYBPC3*, *MYH7*, *TPM1*, *TNNI3*, or *TNNT2* [14]. They demonstrated that the initial trigger of HCM pathogenesis with the *MYH7* mutation is a contractility deficit caused by a sarcomere gene mutation that induces cardiomyocyte hypertrophy and reduced myofibril density, resulting in the reduced force generation capacity observed in the hearts of patients with HCM who have *MYH7* mutations [14]. These findings are consistent with findings from previous studies of the R403Q mutation in *MYH7* using human ventricular muscle cells, which demonstrated reduced enzymatic and contractile activity of R403Q β-cardiac myosin [15]. However, these results should be interpreted with caution because the hearts of patients with HCM have usually undergone extensive cardiac remodeling, such as cardiomyocyte hypertrophy, cardiac fibrosis, lower myofibril density, myofibril disarray, secondary changes in gene expression, and post-translational modifications. These changes can mask the initial insult caused by a mutation, making it difficult to clarify the initial functional changes caused by sarcomere mutations using tissue samples from patients with HCM. Indeed, there are wide variations between studies on the effects of *MYH7* mutations mentioned later.

Cardiomyocytes from patient-derived human induced pluripotent stem cell (hiPSCs) carrying a nonsense mutation in *MYBPC3* (c.2373dupG mutation) showed a significantly lower level of cMyBP-C, less than 50% of normal levels. These cardiomyocytes also exerted significantly less force at the single-cell level, suggesting haploinsufficiency as a mechanism [16]. Mutations in ventricular myosin regulatory light chain (vRLC) encoded by *MYL2* also cause HCM in humans. vRLC wraps around the head-rod junction of the myosin heavy chain, which stabilizes the lever arm of myosin and regulates sarcomere contractility. Porcine cardiac-myosin in which endogenous vRLC was replaced by either N47K or R58Q mutant vRLC is associated with decreased force production and power output compared with wild-type vRLC [17]. Transgenic mice expressing the HCM-causing D166V vRLC mutation in the heart had significantly lower contractile force and abnormally high myofilament calcium sensitivity, resulting in an HCM phenotype [18]. Interestingly, the reduced contractility observed with these vRLC mutations recovered with vRLC phosphorylation via cardiac myosin light chain kinase (cMLCK) [17,18], which prevented the development of HCM in mice [18]. These findings suggest vRLC phosphorylation as a potential therapeutic target for HCM with hypocontractile mutations. Mutations in ventricular myosin essential light chain (vELC) encoded by *MYL3* were also identified in familial HCM despite rare. Most of the HCM-associating mutations in *MYL3* are located in the EF-hand Ca^2+^ binding motifs of the vELC protein [19]. However, functional effects of these mutations on force generation have shown to be different depending on the mutation position in vELC. A57G vELC mutation decreased maximal force generation [19], while E56G vELC mutation increased force generation by redistributing the structure of actomyosin complex toward the strongly bound, force-generation state [20].

### 4.2. Hypercontractile Hypothesis

On the other hand, the hypercontractile hypothesis was proposed based on several studies involving transgenic mice and myosin expressed in vitro in which HCM mutations enhanced contractile activity [5,21,22]. Georgakupoulos et al. showed that mice expressing the familial HCM R403Q mutation in α-myosin heavy chain (αMHC^403+/−^) had enhanced contractility, delayed pressure relaxation, and delayed chamber filling before chamber morphologic or histologic abnormalities appeared in the left ventricle [23]. This is consistent with the observation from a single-molecule mechanical assay study that showed R403Q mutant myosin isolated from αMHC^403+/−^ mice have higher actin-activated ATPase activity, greater average force generation, and faster actin filament sliding velocity [24]. The enhanced contractility observed with R403Q mutant myosin was subsequently observed in studies of R403Q [25,26,27] and other HCM-causing mutations in *MYH7*: D239N [6], H251N [6], R403W [28], R453C [5,26], D906 [29], and L908V [29]. In addition, recent studies have revealed more details about the molecular mechanisms by which HCM-causing mutations enhance contractility based on the ensemble force (*Fe*) equation. The R453C mutation results in a hypercontractile state in the heart muscle by increasing the intrinsic force of β-cardiac myosin [5]. The D239N mutation increases the intrinsic force of β-cardiac myosin [6]. The D251N mutation increases the intrinsic force of β-cardiac myosin and makes more myosin heads available [6]. Recently, Davis et al. identified a significant relationship between the magnitude of tension developed over time and heart growth [30]. Based on this finding, they proposed a tension-based model that can predict HCM and DCM in mice and iPSC-cardiomyocytes associated with any sarcomere gene mutation [30]. This model shows that reduced total tension in cardiomyocytes is strongly correlated with the development of DCM, while increased total tension is correlated with HCM [30].

### 4.3. New Drug Therapy Hypertrophic Obstructive Cardiomyopathy (HOCM)

As mentioned above, there are two competing hypotheses about the initial functional change caused by sarcomere mutations in the development of HCM. However, recent development of a small-molecule inhibitor of sarcomere contractility, mavacamten (MYK-461), provided an unequivocal answer to this question. Excess sarcomere power was shown to be the primary defect in HCM [22]. Mavacamten is a specific inhibitor of cardiac myosin ATPase that lengthens the total cycle time (tc) of the ATPase cycle and also reduces the rate of phosphate release from myosin heads without slowing ADP release [22,31] (Figure 3). This leads to a reduction in duty ratio and ensemble force. It can reduce fractional shortening in adult rat ventricular cardiomyocytes without changing the calcium transient and reduce maximal tension in skinned cardiac muscle fibers without altering pCa_50_ in a dose-dependent manner [25]. In addition to a reduced duty ratio, recent studies revealed that mavacamten exerts its effects primarily by stabilizing the SRX state of β-cardiac myosin [32,33] (Figure 3). Stabilization of the SRX state of cardiac myosin by mavacamten reduces the total number of myosin heads in the sarcomere that are functionally accessible for interaction with actin filaments. Accordingly, mavacamten suppresses cardiac contractility via the combined effects of a lower duty ratio and fewer cross-bridges (Figure 3).

Importantly, chronic mavacamten infusion suppresses the development of the HCM phenotype including ventricular hypertrophy, cardiomyocyte disarray, and myocardial fibrosis in mice models of HCM with the heterozygous R403Q or R453C mutation in *MYH7* [22]. The effect of mavacamten was tested also in a feline HOCM model, which revealed that mavacamten can decrease the left ventricular outflow tract (LVOT) pressure gradient by reducing contractility and eliminating systolic anterior motion of the mitral valve in an exposure-dependent manner [34]. Furthermore, the open-label phase 2 PIONEER-HCM trial examined the effects of mavacamten in HOCM [35]. In this study, mavacamten was orally administered in a total of 21 patients with symptomatic HOCM. Postexercise LVOT gradient and peak oxygen consumption (peakVO_2_) were examined at 12 weeks. Interestingly, mavacamten reduced mean postexercise LVOT gradient from 103 ± 50 mm Hg at baseline to 19 ± 13 mm Hg at 12 weeks (mean change, 89.5 mm Hg [95% confidence interval (CI), 138.3–40.7 mm Hg]) and increased peak VO_2_ by a mean of 3.5 ± 3.3 mL/kg/min (95% CI, 1.2–5.9 mL/kg/ min) [35]. The PIONEER-HCM trial demonstrated that mavacamten had beneficial effects, namely reduction of LVOT obstruction and improvement of exercise capacity in patients with symptomatic HOCM. Based on these results, EXPLORER-HCM, a multi-national, randomized, double-blind, placebo-controlled, phase 3 trial, is currently underway to assess the effects of 30 weeks of treatment with mavacamten on clinical parameters such as peak VO_2_ and New York Heart Association (NYHA) functional classification.

Thus, experimental models and clinical studies support the hypercontractile hypothesis for the pathogenesis of HCM associated with certain sarcomere mutations and mavacamten seems to be an effective drug therapy. However, we need to be especially cautious about types of HCM caused by hypocontractile mutations, in which mavacamten may worsen the natural course. Accordingly, future studies are needed to clarify the initial change in contractility for each HCM mutation.

## 5. Dilated Cardiomyopathy (DCM)

Hereditary DCM can be caused by single point mutations in sarcomere proteins. However, the link between point mutations and clinical phenotypes in DCM is not thoroughly understood in most cases. Recent advances in biochemical, biophysical, stem cell, and gene editing technologies have provided a better understanding of the molecular mechanisms through which the initial insult in DCM (i.e., mutations in a sarcomere protein) induces alterations in cellular organization and contractility, resulting in disease phenotypes. In particular, hiPSC-CMs and genetically modified animals are excellent models because they can capture the initial molecular phenotype that occurs before major compensatory mechanisms mask it.

### 5.1. Genetic Causes of DCM

The most common genetic causes of DCM are truncating variants of titin (TTNtvs), which are observed in approximately 20% of familial or sporadic cases [36]. Titin is a giant sarcomere protein composed of four functionally distinct segments. Titin is anchored to the Z disc through its N-terminal domain and associated with the myosin thick filament through the A-band domain. Titin extends toward the M line of the sarcomere. By connecting myosin filaments to both the Z disc and the M line, titin stabilizes the alignment of the thick filaments. Although numerous rare variants in titin have been reported, most are not pathogenic. Pathogenic TTNtvs observed in patients with DCM are enriched in the titin A-band region [36]. Some A-band TTNtvs produce stable truncated proteins. However, these mutant proteins are not able to assemble with other contractile proteins into well-organized functional sarcomeres, resulting in profound contractile deficits, impaired response to mechanical and β-adrenergic stress, and attenuated growth factor and cell signaling activation [37]. A-band TTNtvs also impair sarcomerogenesis by disrupting mechanical force transmission from β-cardiac myosin II to protocostameres, because titin is a molecular link that couples mechanical force from β-cardiac myosin II with protocostamere-anchored actinin polymerization [38]. One reason why truncating mutations in the I-band region of titin are better tolerated might be that alternative exon splicing in the I-band region mitigates their pathogenicity [37].

Troponin-T, encoded by *TNNT2*, is a subunit of the troponin complex that regulates calcium-dependent interactions between thick and thin filaments. *TNNT2* is one of the most frequently mutated genes in DCM [39]. A deletion of lysine 210 (∆K210) in troponin-T causes early-onset DCM [40]. Clippinger et al. demonstrated that the ∆K210 mutation shifts the equilibrium positioning of tropomyosin by reducing the number of strongly bound myosin cross-bridges via decreases in the fraction of thin filaments in the open, weakly bound, and strongly bound states and via increases in the fraction of thin filaments in the blocked and close states [40]. They also demonstrated that ∆K210 cardiomyocytes have defects in mechanosensing that may prevent adaptive responses in their contractility and structure in response to changes in the mechanical environment such as disease-related stiffening of cardiac tissue [40].

Mutations in vRLC, in which aspartic acid at position 94 is replaced by alanine (D94A), also result in DCM [41]. D94A-vRLC is associated with reduced incorporation into the myosin heavy chain due to impaired binding to the myosin heavy chain, resulting in less maximal tension in D94A-reconstituted skinned porcine papillary muscle strips compared with wild-type muscle strips [41]. A small-angle X-ray diffraction study showed that D94A-vRLC-reconstituted myosin is hypocontractile by demonstrating the repositioning of the D94A cross-bridge mass toward the thick filament backbone [42].

Certain *MYH7* mutations are known as DCM-causing mutations [43]. The DCM-causing mutations S531P and F764L are associated with contractile deficits [43], in contrast to the HCM-causing *MHH7* mutations. All DCM-causing sarcomere mutations are associated with hypocontractile sarcomeres, which is consistent with the tension-based model [30].

### 5.2. Challenges in the Development of New Drug Types for Systolic Heart Failure

Patients with advanced systolic heart failure continue to have poor prognosis. Inotropes are crucial in managing patients with heart failure who have reduced cardiac output and poor end-organ perfusion. However, the use of traditional inotropes remains controversial because inotropes improve hemodynamic status at the expense of an increased risk of mortality. Adverse effects observed with traditional inotropes are thought to be caused by increased intracellular calcium concentration. Thus, the development of inotropic agents that can improve cardiac function without adverse clinical events had been a long-cherished desire in the field of cardiovascular medicine.

#### 5.2.1. Omecamtiv Mecarbil (OM)

Omecamtiv mecarbil (OM), a recently developed direct sarcomere activator, can exert inotropic effects by directly enhancing sarcomere contractility without affecting intracellular calcium concentration [44]. OM is a small molecule compound (molecular weight 401.43 g/mol) that selectively and allosterically activates cardiac myosin by binding to a narrow cleft between the N-terminal 25-kDa domain and the 50-kDa domain of the motor domain [44]. OM increases the equilibrium constant of the ATP hydrolysis step (M-ATP→M-ADP-Pi) by 2.5-fold, shifting toward product formation (actomyosin-ADP-Pi) in the active site [45] (Figure 4).

OM also increases phosphate release from myosin by four-fold [44,45] (Figure 4). This mechanism is supported by the structural analysis of human β-cardiac myosin and OM [46], which revealed that OM stabilizes the links responsible for the rotation of the lever arm into the pre-power stroke conformation by binding with key residues in the 25-kDa domain, relay helix, SH1 helix, and converter domain. It also revealed that OM binding results in a twist of the β5, β6, and β7 strands of the seven-stranded β-sheet in the transducer region, which may be involved in γ-phosphate release [46] (Figure 4). Thus, mechanistically OM acts as an inotrope by increasing the concentration of the actomyosin-ADP-Pi intermediate and accelerating the release of phosphate from myosin heads, which results in a higher duty ratio and more myosin heads in the force-generating state (Figure 4). In cultured adult rat cardiomyocytes, OM increases fractional shortening and the duration of contraction without affecting the calcium transient [44]. Importantly, its effects are independent of β-adrenergic signaling because OM can enhance contractility even in the presence of carvedilol, a β-adrenergic blocker commonly used in patients with heart failure [26]. In a canine heart failure model, OM increases stroke volume and cardiac output and lowers heart rate. In contrast to the β-adrenergic agonist dobutamine, which enhances contractility by increasing dP/dt and shortening systolic ejection time (SET), OM does not change dP/dt and increases SET. Thus, OM is expected to provide a new therapeutic approach for systolic heart failure. Clinical trials in patients with systolic heart failure are underway.

The Chronic Oral Study of Myosin Activation to Increase Contractility in Heart Failure (COSMIC-HF trial) is a phase 2, pharmacokinetic, randomized, double-blind, placebo-controlled study. It was conducted at 87 sites in 13 countries [47]. This trial enrolled 448 patients with HFrEF (age 18–85 years, NYHA class II or III disease, left ventricular ejection fraction ≤ 40%, plasma NT-proBNP ≥ 200 pg/mL) treated with stable, optimum pharmacological therapy for at least four weeks. Patients were randomly assigned in equal ratios to the fixed-dose group (25 mg twice daily), the pharmacokinetic (PK)-titration group (25 mg twice daily titrated to 50 mg twice daily guided by PK), or the placebo group for 20 weeks. Mean maximum concentration of OM at 12 weeks was 200 ± 71 ng/mL in the fixed-dose group and 318 ± 129 ng/mL in the PK group. At 20 weeks, both the fixed-dose group and the PK group had significant increases in ejection time and systolic volume and a significant decrease in NT-proBNP plasma levels compared with the placebo group. Furthermore, the PK group had a significant reduction in left ventricular end-systolic dimension, LV end-diastolic dimension, and heart rate. Importantly, the incidence of clinical adverse events in patients on OM was comparable to the incidence in the placebo group. The observed effects of OM in this study were quite different from those of traditional inotropic agents, indicating that direct sarcomere activation can not only improve cardiac function, but also reduce ventricular wall stress, suppress sympathetic activation, and promote favorable ventricular remodeling. Thus, the COSMIC-HF trial showed that OM dosing guided by PK resulted in improved cardiac function and a reduction in LV diameter without increasing clinical adverse events in patients with stable HFrEF. To assess the safety and efficacy of OM in patients with chronic HFrEF receiving standard therapy for heart failure, the Global Approach to Lowering Adverse Cardiac Outcomes Through Improving Contractility in Heart Failure (GALACTIC-HF trial) is underway. The GALACTIC-HF trial is a phase 3, randomized, double-blind, placebo-controlled, multicenter study that started in January 2017 with a target completion date of January 2021.

#### 5.2.2. Other Target Molecules for Direct Sarcomere Modulators

Although OM is a new promising inotropic agent for the treatment of HFrEF, clinicians need to be aware of the risk of myocardial ischemia with overdosing of OM [48,49]. Thus, it is necessary to develop other types of direct sarcomere activators as well. Earlier drugs that directly activate sarcomere contractility target myosin and the troponin complex [50]. However, other sarcomere proteins have also been proposed as therapeutic targets for correcting contractile dysfunction in heart failure. Cardiac myosin binding protein C (cMyBP-C), which interacts with myosin S2 and actin, is an attractive target molecule because its phosphorylation enhances force generation and force relaxation by accelerating the rate of cross-bridge recruitment and detachment, respectively. Most cMyBP-C molecules are phosphorylated in normal human hearts but mostly dephosphorylated in failing hearts [51]. vRLC is also an attractive target molecule for direct sarcomere modulation because its level of phosphorylation is well known as a physiological regulator of sarcomere contractility that is reduced in failing hearts [52].

## 6. Conclusions

Genetic studies have identified increasing numbers of sarcomere gene mutations that cause cardiomyopathies. Although recent advances in genetic screening have made it possible to identify sarcomere mutations at an early stage before the onset of cardiomyopathy in healthy family members (mutation carriers) of patients, no therapies to prevent cardiomyopathies are currently available. Mavacamten and OM, recently developed direct sarcomere modulators, have demonstrated significant therapeutic effects in HOCM and DCM, which were not observed with traditional therapies. However, the ideal therapy may depend on the nature of each mutation. Recent technological advances have started to clarify the initial functional insults through which specific gene mutations in sarcomeres cause contractile dysfunction. This knowledge is necessary for the development of mutation-specific therapies to prevent or blunt the progression of cardiomyopathies. Further development of direct sarcomere modulators that can correct sarcomere function according to mutation-specific mechanisms is expected in the future.

## Figures and Tables

**Figure 1 ijms-21-00226-f001:**
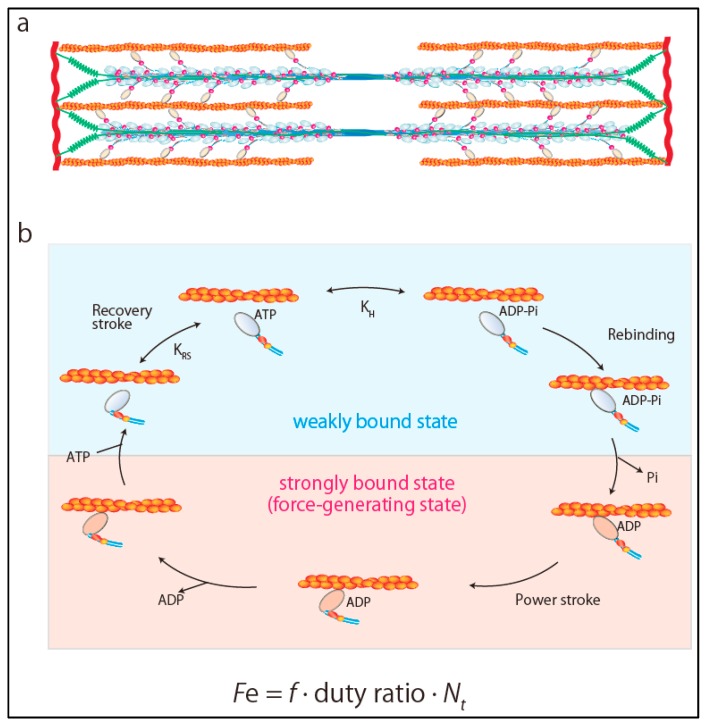
Ensemble force of sarcomere. (**a**) Structure of sarcomere. (**b**) Chemomechanical cycle of actomyosin. Ensemble force is determined as *Fe* = *f* × duty ratio × *N_t_*.

**Figure 2 ijms-21-00226-f002:**
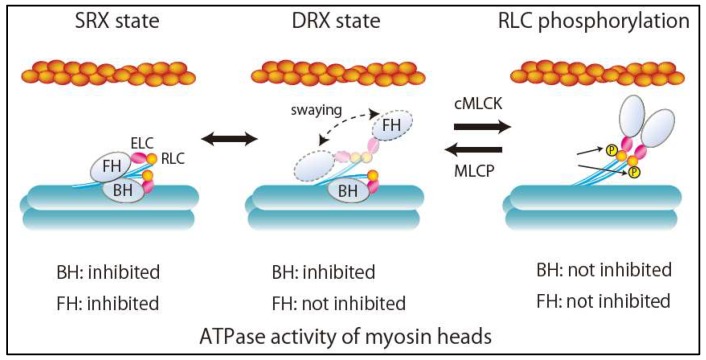
Structures and ATPase activity of myosin heads during relaxation states. SRX, super relaxation; DRX, disordered relaxation; FH, free head; BH, blocked head; ELC, myosin essential light chain; RLC, myosin regulatory light chain; cMLCK, cardiac myosin light chain kinase; MLCP, myosin light chain phosphatase.

**Figure 3 ijms-21-00226-f003:**
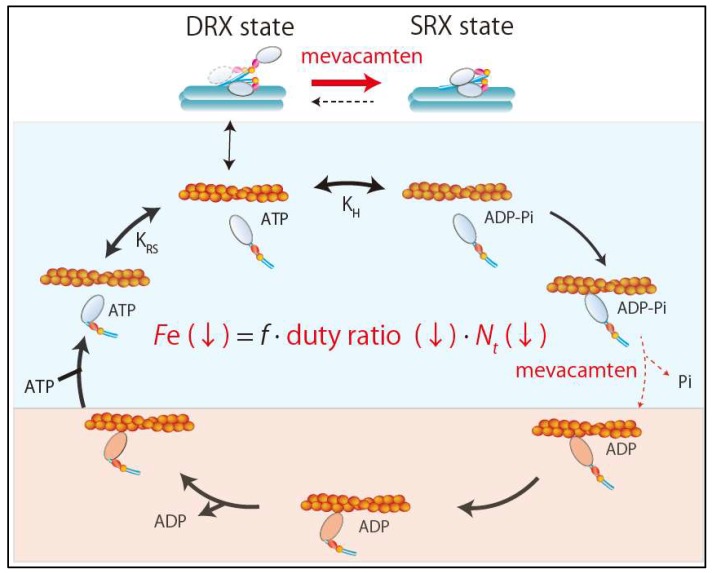
Effects of mevacamten. Mevacamten reduces the ensemble force by combination of reductions in the duty ratio and the number of total number of functionally available myosin heads. SRX, super relaxation; DRX, disordered relaxation.

**Figure 4 ijms-21-00226-f004:**
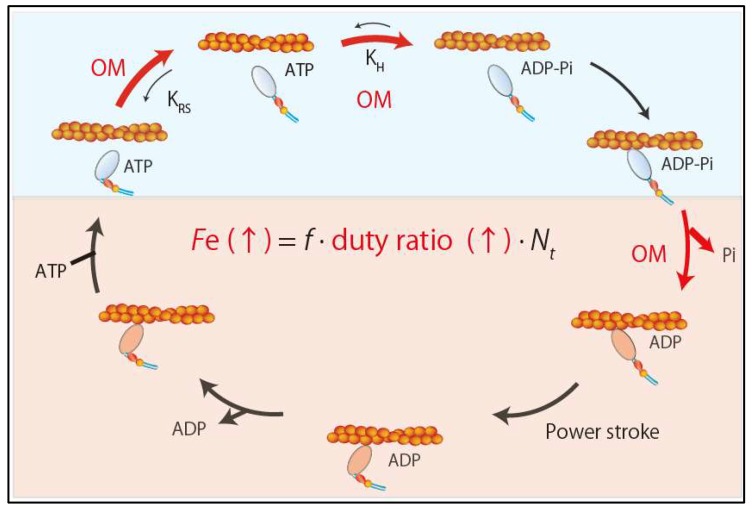
Effects of omecamtiv mecarbil. Omecamtiv mecarbil increases the ensemble force by increasing duty ratio.

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
