# Peer review of "Direct Sarcomere Modulators Are Promising New Treatments for Cardiomyopathies"

_ijms, 2019, doi:10.3390/ijms21010226_

Round 1

Reviewer 1 Report

This is a good and timely review on the causal links between mutations to sarcomeric proteins and cardiac syndromes, particularly hypertrophic cardiomyopathy (HCM) and dilated cardiomyopathy (DCM). Recent advances have shown that HCM is due to hypercontractility-inducing mutations, mainly to myosin heavy chains, and contractility inhibitors are potentially useful to treat HCM patients. Conversely, DCM is mainly due to hypocontractility, and calcium-independent myosin activators could be novel therapeutic tools to treat DCM patients.

The review is, on average, well-presented, balanced, timely and adequate. However, there is some room for improvement. A general issue is that the transition between the ensemble power and sarcomere power output section (2) and the one on the states of heart myosin (3) is too abrupt. I understand that some of this should be common knowledge for the reader, but the paper moves from structural physiology (lines 56-79) to a highly detailed discussion on the molecular detail of myosin conformation inside the sarcomere without any introduction of the role of myosin in the sarcomere, its force-generating mechanism, etc. At the risk of repeating concepts that appear in textbooks, I think a short (30 lines or less) discussion on the topic between current sections 2 and 3 would be useful. The rest of the review is excellent.

Several typos and spelling mistakes are present throughout and need to be corrected. The ones I caught:

1 “Recovery storke” should read “recovery stroke”. Also, make sure Recbinding is correct, because I do not recognize this wording. 2, RCL should read “RLC”. Also, in the intermediate state, the shaded, “swayed” head should be lighter or colorless. Line 208, confidential should be “confidence”. Line 257, position 94 should be “position 49” Please revise that all the mutations described are properly numbered.

Author Response

Thank you very much for your comments. Our response to each of your comments and the changes made to our manuscript are listed in the text as follows.

Comment 1.

The paper moves from structural physiology to a highly detailed discussion on the molecular detail of myosin conformation inside the sarcomere without any introduction of the role of myosin in the sarcomere, its force-generating mechanism, etc. At the risk of repeating concepts that appear in textbooks, I think a short (30 lines or less) discussion on the topic between current sections 2 and 3 would be useful.

Answer 1.

Thank you for helpful and productive advices. We completely agreed with it and added a short discussion between current sections 2 and 3 (Line 90-109). We also added a short introduction about the force-generating mechanism of the sarcomere in the section 2 (Line 65-72).

Comment 2.

Several typos and spelling mistakes.

Answer 2.

“Recovery storke” and “Recbinding” were corrected as “recovery stroke” and “Rebinding”, respectively, in figure 1. In figure 2, “RCL” was corrected as “RLC”. Also, I made the colors of the shaded, “swayed” heads lighter. “confidential” was corrected as “confidence” (Line 246). I am sorry but “position 94” is correct. So, I corrected “D49A” as “D94A” (Line 295).

Reviewer 2 Report

This is an excellent review article that contains original and important insights on the genetic cardiomyopathies and on novel pharmacological approaches to treat them.

It is well organized and written.

Author Response

Thank you for your favorable comments.

Reviewer 3 Report

This review discusses mutations in various sarcomere genes as applicable to hypertrophic cardiomyopathy (HCM) and dilated cardiomyopathy (DCM). Excellent background is provided on cardiac power output and relation to cardiac disease is explained well. The author has also discussed therapeutic efforts in treating HCM and DCM.

In the section on HCM, hypo- and hypercontractile hypothesis is discussed. Although myosin heavy chain mutations are discussed, myosin light chains are ignored. Mutations in MYL2/3 have been identified in hypertrophy as well. Admittedly, mutations in myosin essential and regulatory light chain are rare (<5%). Nevertheless, discussion about light chains could be added to the review for completeness.

The text needs to be proofread and the several typographical errors corrected. For example, the word "title" is present in the article title. Several other errors were also present.

Author Response

Thank you very much for your comments. Our response to each of your comments and the changes made to our manuscript are listed in the text as follows.

Comment 1.

Discussion about light chains could be added to the review for completeness.

Answer 1.

Thank you for helpful and productive advices. We completely agreed with it and added the discussion about myosin essential light chain in the section 4 (Line 185-191).

Comment 2.

The several typographical errors.

Answer 2.

The word “title” was removed from the article title. “Recovery storke” and “Recbinding” were corrected as “recovery stroke” and “Rebinding”, respectively, in figure 1. In figure 2, “RCL” was corrected as “RLC”. Also, I made the colors of the shaded, “swayed” heads lighter. “confidential” was corrected as “confidence” (Line 246). “D49A” was corrected as “D94A” (Line 295).